# Energy Reconstruction and Calibration of the MicroBooNE LArTPC

Richard Diurba  on behalf of the MicroBooNE Collaboration

Laboratory for High Energy Physics, University of Bern, 3012 Bern, Switzerland; microboone_info@fnal.gov

**Abstract:** MicroBooNE uses a liquid argon time projection chamber (LArTPC) for simultaneous tracking and calorimetry. Neutrino oscillation experiments plan to use LArTPCs over the next several decades. A challenge for these current and future experiments lies in characterizing detector performance and reconstruction capabilities with thorough associated systematic uncertainties. This work includes updates related to LArTPC detector physics challenges by reviewing MicroBooNE's recent publications on calorimetry and its applications. Highlights include discussions on signal processing, calorimetric calibration, and particle identification.

**Keywords:** liquid argon detectors; liquid argon calibration; signal processing; particle identification

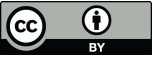

## 1. Introduction

MicroBooNE is an 85 metric tonne liquid argon time projection chamber (LArTPC) that operated from 2015 to 2021 at the Fermi National Accelerator Laboratory (Fermilab) [1]. The detector sits roughly 500 m on-axis from the Fermilab Booster Neutrino Beam and expects an on-axis flux of neutrinos with energies of approximately 0.2 to 2 GeV [2]. LArTPCs operate by drifting ionized electrons from particles traversing the detector in an electric field to readout panels. The drifting of ionized electrons onto, in the case of MicroBooNE, arrays of wire planes, allows for simultaneous tracking and calorimetry of charged particles traversing the argon. For simulation and data, it uses the LArSoft software kit for decoding data packets, simulating the detector, and processing the reconstruction [3,4]. The energy deposition of particles in the detector is simulated with GEANT4 [5–7]. The simulation event generator for neutrino interactions is GENIE [8–11]. For cosmic ray muons, it is CORSIKA [12].

These proceedings intend to review the work of MicroBooNE over the years to develop a robust and thorough signal processing and calibration scheme for precision hit-by-hit calorimetry. The article will finish with examples of applications of how MicroBooNE exploits its precision calorimetry data for higher-level reconstruction, such as particle identification and shower clustering.

## 2. Signal Processing

The LArTPC of MicroBooNE has three wire planes. Two wires planes, Plane 0 and Plane 1, have angles ±60 degrees from the third plane, Plane 2. Plane 0 and Plane 1 measure ionized electrons in the argon using the induced signal of electrons traveling towards and away from these induction wire planes. The drifting electrons end on Plane 2, which reads a signal from the ionized electrons by collecting them on the wire. The combination of the two induction planes and one collection wire plane allows for three-dimensional tracking of the ionized electrons that remain from the passage of a charged particle in the argon. The electric field at MicroBooNE operates at 273 V/cm [1].

The wire planes undergo noise filtering and then two-dimensional deconvolution to eliminate effects from the electronics and sharpen signals to isolated wires. Figure 1 shows a data neutrino event candidate going through each stage of signal processing

from raw waveforms, noise filtering, and deconvolution [13,14]. Concurrently with noise filtering, the electronic response calibration is applied. With the implementation of electronics response filtering and calibration, the blue bands in Figure 1 disappear, eliminating extraneous reconstructed hits surrounding the candidate neutrino interaction vertex. Without removing these extraneous signals, the reconstruction may incorrectly identify extra tracks and incorrect vertexes. The 2D deconvolution intends to sharpen signals and extract the ionization charge from the smearing of the electronics response. The process includes a Fourier transformation and a low-pass filter [13,14]. Downstream reconstruction can then form hits from these waveforms. An example would be to use Gaussian functions to extract the full charge [15].

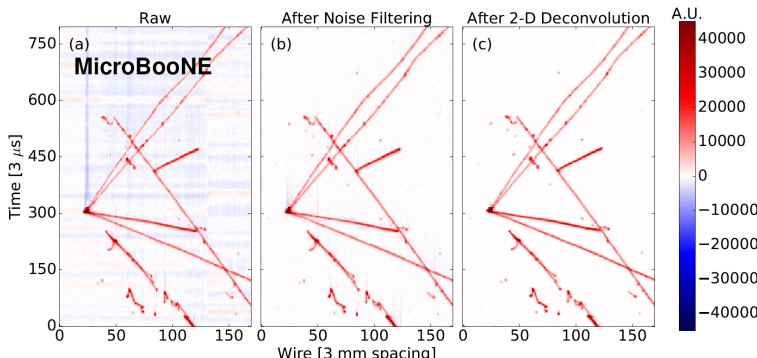

**Figure 1.** Candidate neutrino interaction event display from MicroBooNE data through each stage of signal processing from raw data (**a**), noise filtering (**b**), and finally the event with both noise filtering and 2D deconovolution (**c**). Figure was taken from [14].

The wire response simulated depends on the liquid argon property values used. For example, the diffusion of drift electrons alters how the simulation generates simulated waveforms. If the diffusion constants between data and simulation differ, then the waveforms between simulation and data may vary enough to propagate discrepancies to higher-level reconstruction variables. A method developed by MicroBooNE to factor in differences in waveforms in simulation and data is to modify the waveforms as a function of track variables [16]. The process is twofold. First, ratios of the hit charge and hit width of reconstructed waveforms are made from hits on cosmic muon tracks as a function of drift distance (X), height (Y), distance across the length of the detector (Z), and angles ($\theta_{XZ}$, $\theta_{YZ}$). These hits come from a Gaussian fit to regions of interest on the waveform [16].

Figure 2 shows the ratio for all three wire planes as a function of position across the drift distance (x). The trajectories of the cosmic ray muons simulated were taken from real reconstructed data cosimc ray muons, and all other elements of the simulated muons come from CORSKIA [12].

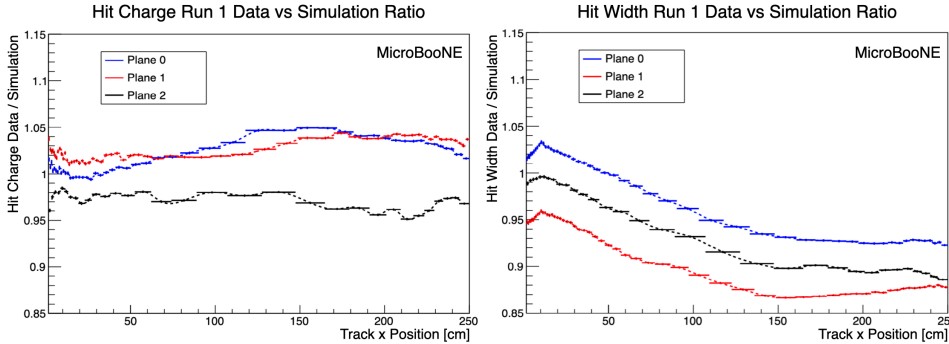

**Figure 2.** Plots of the ratios between data and simulation as a function of drift distance (X). The anode is approximately at x = 0. Images come from [16].

Second, these ratios go into a function that creates variation simulation samples, one made for each trajectory variable (X, YZ, $\theta_{XZ}$, $\theta_{YZ}$) for a total of four samples. The varied samples change the simulated waveforms associated with simulated neutrino interactions, which will be independent of the cosmic ray muon samples used to generate the ratios. The waveforms in these varied simulation samples are modulated by the hit charge and hit width ratios measured between data and simulation for the various track variables. In the case of overlapping hits, only the portion not related to the cosmic ray muon is modulated. The process begins by creating a scale factor. The scale factor is the ratio as a function of track position weighted by the amount of energy deposited in the simulation. This new weighted ratio value for the associated hit width and hit charge rescales the varied hit width and hit charge. These new, altered hit widths and hit charges are then fed into a Gaussian function, and the waveform associated with the hit is scaled relative to the unaltered simulated hit width and hit charge. Equation (1) shows the reweighting function as a function of the drift time in terms of the mean time (t), width ($\sigma$), and charge (Q) between the original hit ($\sigma$, Q) and the reweighted hit ($\sigma'$, Q') for times associated with the waveform being varied ($t_j$).

$$w = \frac{\sum_j \frac{Q'}{\sqrt{2\pi\sigma'^2}} \exp\left(-\frac{(t-t_j)^2}{2\sigma'^2}\right)}{\sum_j \frac{Q}{\sqrt{2\pi\sigma^2}} \exp\left(-\frac{(t-t_j)^2}{2\sigma^2}\right)} \tag{1}$$

Figure 3 reveals two examples of the wire modification on both a single hit and overlapping hits [16]. These varied simulation samples, four wire modification samples in total, were then used to evaluate the detector-related systematic uncertainties for analyses. Examples include the following publications [17–20].

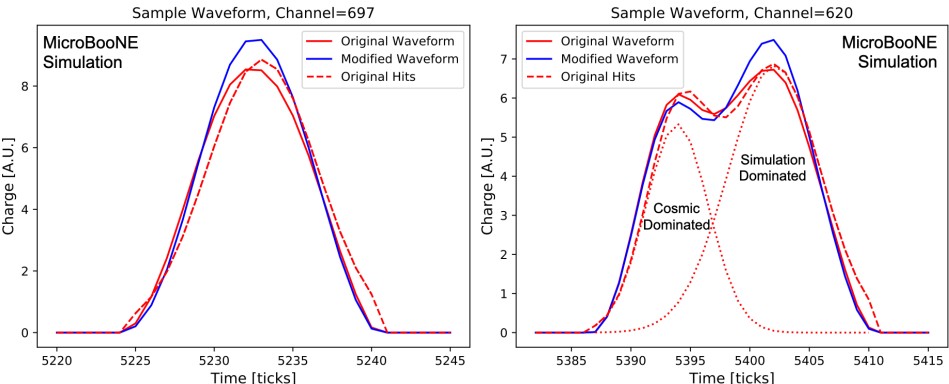

**Figure 3.** Examples of the original and modified waveforms for a single hit (**left**) and two concurrent hits from a cosmic ray muon overlaid in simulation and a simulated physics interaction (**right**). Only the waveform from the simulated physics event was modulated to form the new hit in the varied sample. The images come from [16].

## 3. Calibration of TPC Calorimetry

The precision of the energy reconstructed in data and simulation depends on a thorough and robust calibration of the amount of energy deposited per unit length. The need for detailed calorimetry calibration is especially true for LArTPCs, since the ability to measure individual energy deposits at sub-millimeter resolution is a proposed advantage of using this detector technology over others. In the case of MicroBooNE, the scheme calibrates the hit charge measured per unit length (dQ/dx) as a function of position and time. It then calibrates the energy scale using calorimetry from selected stopping muon candidates [21]. The two-step calibration has been adapted from the calibration schemes in MINOS [22]. In MicroBooNE's calibration process, each correction measured is used to generate the next set of dQ/dx corrections.

Since MicroBooNE operates on the surface, the cosmic ray muon flux is so significant that abundances of ionized electrons and positively charged argon ions perpetually exist in the detector. These positively charged argon ions move more slowly through the argon, and therefore build up on the edges of the TPC, pushing the ionized electron away from the detector faces [23]. This distortion is known as the space charge effect and leads to the stretching and squeezing of signals. This effect has been measured by studying the positional offset of cosmic ray muons entering and exiting the detector [23]. These measurements were then verified using a UV-laser [23].

It is essential to have a consistent standard candle of tracks, so the corrections of dQ/dx address detector effects and not the physics of interactions and stopping particles. For MicroBooNE, non-stopping cosmic ray muons are used that are anode-cathode crossing. These muons cross the whole drift volume. The arrival time can be ascertained by the hit closest to the anode. These tracks were identified in simulations and data with the Pandora reconstruction package [24,25].

A selected cathode–anode crossing muon track for dQ/dx calibration must have a track length between 250 and 270 cm. It must also have an angle relative to the drift distance and detector length ($\theta_{XZ}$) of less than 75 degrees and an angle relative to the detector height and length ($\theta_{YZ}$) less than 80 degrees. These cathode–anode crossing track samples calibrate each day of data-taking, and therefore are organized by the day the detector collected the event.

These cosmic ray muons samples are used to smear the dQ/dx calibration as a function of position. The smearing function (C) for a generic position variable (i), such as YZ or X, is seen in Equation (2) in terms of the global median dQ/dx and the local dQ/dx.

$$C_i = \frac{dQ/dx_{i,global}}{dQ/dx_{i,local}} \tag{2}$$

Equation (2) is first used in terms of the detector height and length (YZ) to form $C_{YZ}$. The event sample statistics in data for YZ are shown in Figure 4. The corrections in YZ aim to address effects, such as unresponsive channels, space charge effect distortions in YZ, and electronics response. Then, the same process is used for $C_X$ to calibrates as a function of drift distance (X), which corrects for attenuation due to electronegative impurities, diffusion effects, and remaining space charge effect corrections in the horizontal direction. The final calibration of dQ/dx aims to fix time-dependent differences between data-taking days [21] (Equation (3)).

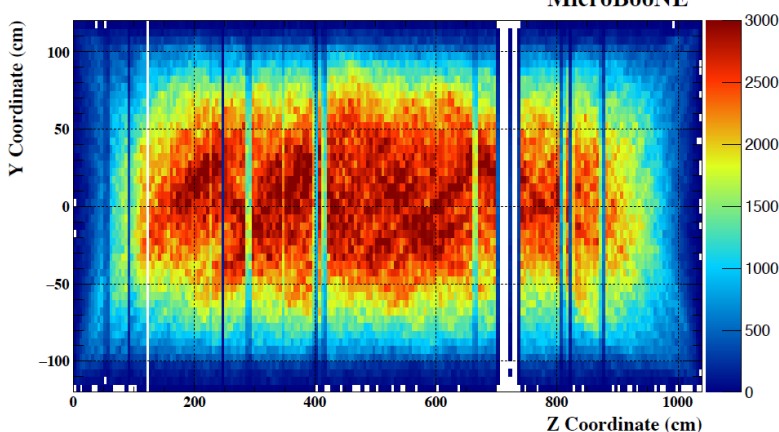

**Figure 4.** Display from data of number of hits measured as a function of position of the detector height (Y) and length (Z). The image is from [21].

$$C(t) = \frac{dQ/dx_{ref.}}{dQ/dx_{global\,median\,xyzcorr.}(t)} \tag{3}$$

The calibrated dQ/dx is finally shown in Equation (4).

$$dQ/dx_{calib.} = dQ/dx \cdot C_X \cdot C_{YZ} \cdot C(t) \tag{4}$$

With the dQ/dx calibrated, the next step is to measure the energy scale. The energy scale measurement starts by selecting a dE/dx model for liquid argon and then measuring a conversion scale, or gain, from ADC values from the TPC electronics to the number of electrons ionized in the hit. MicroBooNE, for calibration, uses the modified Box model [26]. The gain is measured using a sample of neutrino-induced stopping muons with a track length of at least 150 cm and angular cuts identical to those used for the dQ/dx calibration. Like with dQ/dx, Pandora is used via the reconstruction package [24,25]. Most probable values are found for bins of residual range using a Landau–Gaussian fitter [27]. The calorimetric most probable value for dQ/dx in a bin is compared to the Landau–Vavilov predicted value in the region between 250 and 450 MeV. The gain value is found by minimizing the $\chi^2$ between a sample of stopping cosmic ray muons and the expectation from the Landau–Vavilov theory [21,28]. Figure 5 shows the dE/dx of a stopping muon after measuring the gain value. Even outside the kinetic energy range used for calibration, there appears good agreement within uncertainties between the fitted data and the predicted values from Landau–Vavilov. The conversion from residual range to kinetic energy is accomplished using the continuous slowing down approximation table for muons [29]. The gain value measured for the modified Box model extracted from the $\chi^2$ fit is then used as a global value for the data set and works as the final calibration step to convert electronics response to units of energy deposited [21].

Figure 6 evaluates the difference between two methods of assessing the total energy of neutrino-induced stopping muons, the hit-by-hit calibrated calorimetry from the collection plane, and the track length of the muon. The difference between the two methods in data is around 2%, which is near the predicted difference from simulation (1%) and considered a sufficient closure test for calibration [21].

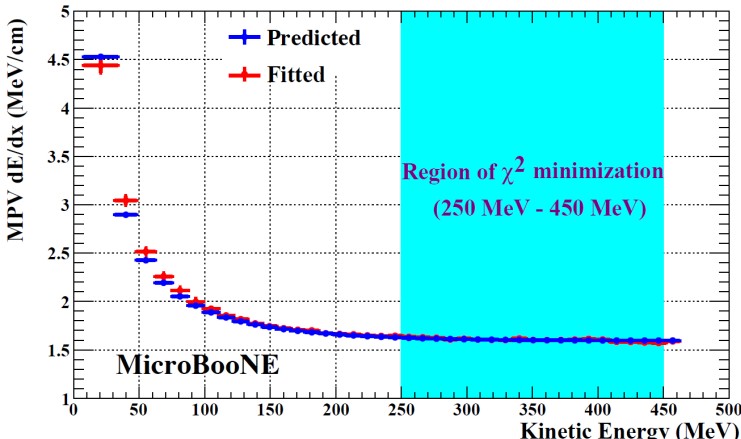

**Figure 5.** Energy deposited (dE/dx) as a function of the cosmic ray muon's kinetic energy for the collection plane calorimetry. The red represents data from 2016 with the best fit of the gain value, and the blue represents the expectation from [28]. The figure was originally seen in [21].

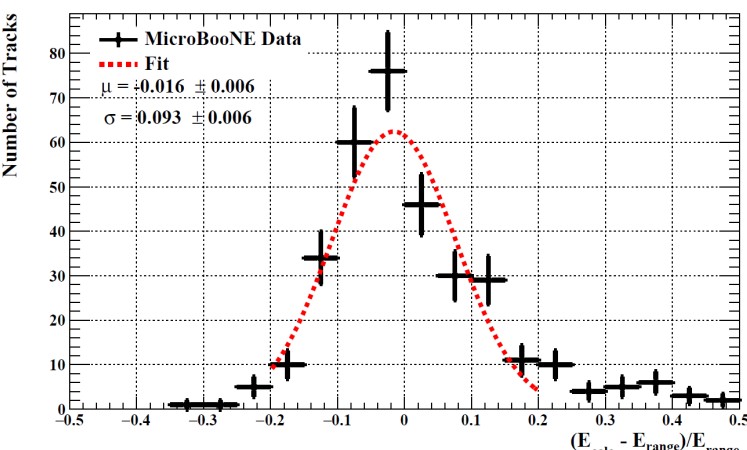

**Figure 6.** Comparison between the total energy of stopping muons measured by hit-by-hit calorimetry and the energy calculated from the track range. Taken from [21].

## 4. Example Applications of Higher-Level Reconstruction Using Calorimetry

MicroBooNE has developed a wide range of reconstruction techniques using calorimetry information. A log-likelihood ratio metric designed to separate reconstructed muons from protons serves as an example. In this method, a log-likelihood metric is from probing template probability density functions of dE/dx in slices of residual range of a muon or proton for each hit in each plane in the last 30 cm of the track [30]. Figure 7 shows the data to simulation comparison as separated by particle type in the simulation, with $-1$ hypothesizing a proton and 1 hypothesizing a muon.

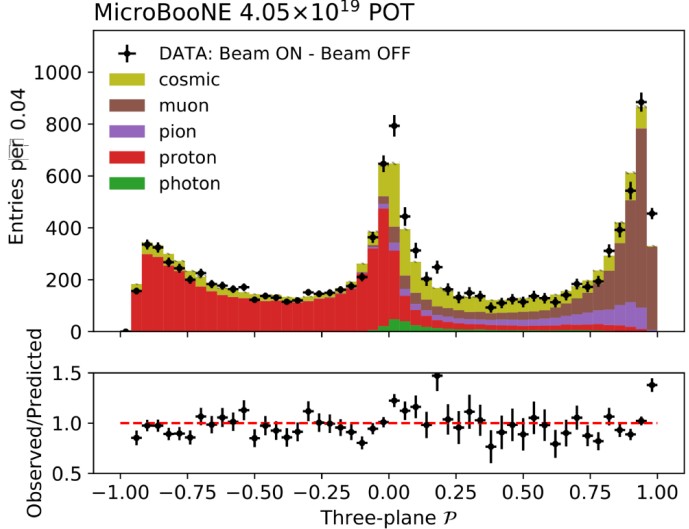

**Figure 7.** Distribution of data and simulation log-likelihood ratio measured as the difference between beam on samples and beam off samples to eliminate the cosmic background. With the Poisson uncertainties on the data distribution, the simulation and data appear within agreement. The image is from [30].

Another recent highlight is the publication of shower reconstruction using deep-learning methods with a SparseSSNet [31]. As a LArTPC, MicroBooNE has the unique capability of identifying photon-induced showers from electron-induced showers. Some analyses used a Kalman filter to accomplish this [17,32]. In tandem, a deep-learning selection using SparseSSNet can identify and isolate electromagnetic showers. The isolated hits of the shower form the total calorimetric energy measured for the event. Figures 8 and 9

reveal the total energy reconstructed with Michel electrons and neutral pions. There is good agreement between data and simulation with $\chi^2/d.o.f$ of 0.61 and 0.98, respectively.

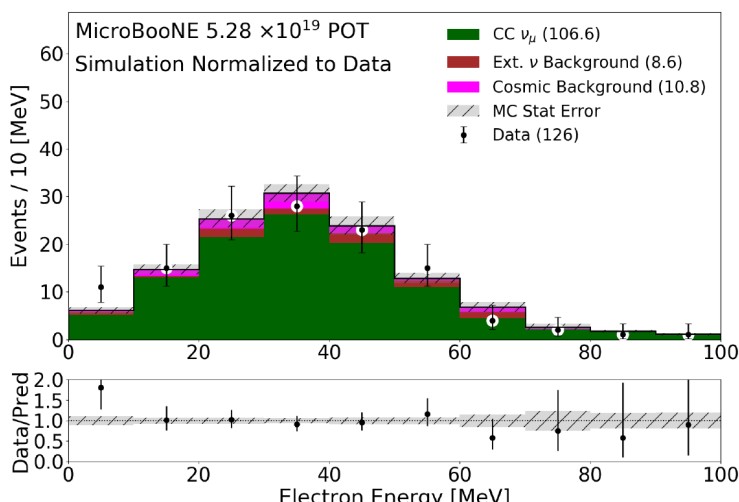

**Figure 8.** Total electron energy reconstructed from selected stopping muons using charged current (CC) muon neutrino events. Statistics for the simulation are scaled to the beam protons on target used in the data distribution. Plot from [31].

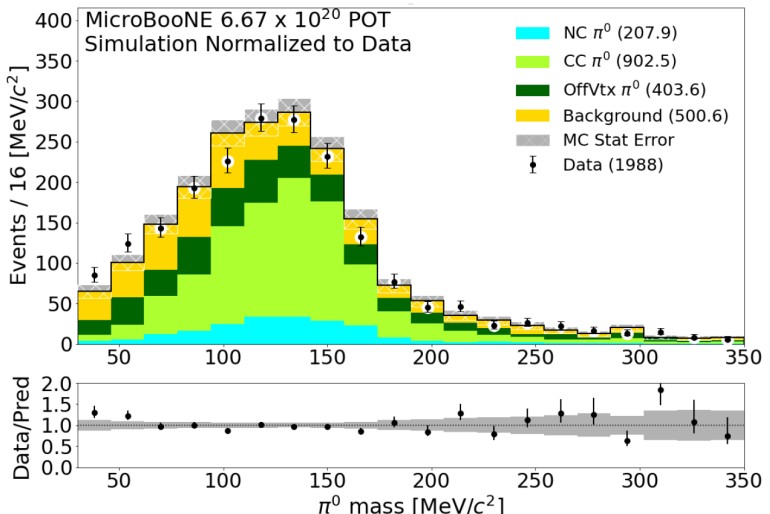

**Figure 9.** Total reconstructed neutral pion mass from charged current (CC) events, neutral current (NC) events, and neutrino interaction events off vertex (OffVtx). Statistics in simulation scaled to beam protons on target used for data distribution. Figure originally from [31].

## 5. Conclusions

This paper summarized recent publications from MicroBooNE. Highlights focused on extracting charge from waveforms, calibrating dE/dx using TPC hits, and applying the calorimetry to reconstruct shower energies and separate proton tracks from muon tracks. MicroBooNE has collected nearly six years of neutrino data and aims to continue developing techniques and applications for LArTPC reconstruction. As an example, techniques discussed were used in publications of cross section results [33,34] and publications searching for anomalous excesses of electron-like neutrino events [17–20].

**Funding:** This document was prepared by the MicroBooNE collaboration using the resources of the Fermi National Accelerator Laboratory (Fermilab), a U.S. Department of Energy, Office of Science, HEP User Facility. Fermilab is managed by Fermi Research Alliance, LLC (FRA), acting under contract number DE-AC02-07CH11359. MicroBooNE is supported by the following: the U.S. Department

of Energy, Office of Science, Offices of High Energy Physics and Nuclear Physics; the U.S. National Science Foundation; the Swiss National Science Foundation; the Science and Technology Facilities Council (STFC), part of the United Kingdom Research and Innovation; the Royal Society (United Kingdom); and The European Union's Horizon 2020 Marie Sklodowska–Curie Actions. Additional support for the laser calibration system and cosmic ray tagger was provided by the Albert Einstein Center for Fundamental Physics, Bern, Switzerland.

**Data Availability Statement:** Not applicable.

**Acknowledgments:** We also acknowledge the contributions of technical and scientific staff to the design, construction, and operation of the MicroBooNE detector; and the contributions of past collaborators to the development of MicroBooNE analyses, without whom this work would not have been possible.

**Conflicts of Interest:** The author declares no conflict of interest.

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
