# Peer review of "Energy Reconstruction and Calibration of the MicroBooNE LArTPC"

_instruments, doi:10.3390/instruments6030030_

Round 1
Reviewer 1 Report
There is a good overview of the MicroBooNE experiment techniques for energy reconstruction and calibration.
Please find some minor corrections proposed below.
Lines 28-29 and Figure 2:
The planes in fig.2 are defined as "Plane 0", "Plane 1" and "Plane 2", while in the text (lines 28-29) they are mentioned as U, V and Z.
Please clarify the association between 0,1,2 and U,V,Z (e.g. in lines 28-29 or in figure caption).
Figure 3:
It is mentioned in the caption that the modified hit is shown, while the original hits only are shown as well as the original and modified waveforms as follows from the legend.
If my understanding is correct, the more clear beginning of the caption might be: "Examples of original and modified waveforms and original hit time distributions for single hit (left) and two ...."
Line 74:
The mean time t0 is not present in the following formula 1.
Line 75:
Typo: "sigma'" should be corrected in Latex.
Line 92:
Possibly omitted verb.
As a LArTPC on the surface --> As a LArTPC stays on the surface
Equation 2:
What does subscript "i" corresponds to?
Should be clarified.
Equation 4:
The Cx and Cyz are not defined.
If I understood correctly, they should be somehow associated with eq.2: e.g. C_{i,YZ} and C_{i,X}.
Figure 6 caption:
and the track range --> and the energy calculated from the track range
Line 137:
Typo: Landua --> Landau
Figure 8:
The legend notations should be clarified in the caption (or in the text body): what are "CC" and numbers shown in parentheses (weighted numbers of events in mc?).
Figure 9:
The legend notations should be clarified in the caption (or in the text body): what are "NC", "CC", "OffVtx", Background and numbers shown in parentheses.
Author Response
- Lines 28-29 and Figure 2: The planes in fig.2 are defined as "Plane 0", "Plane 1" and "Plane 2", while in the text (lines 28-29) they are mentioned as U, V and Z. Please clarify the association between 0,1,2 and U,V,Z (e.g. in lines 28-29 or in figure caption).
- Now says, "The LArTPC of MicroBooNE has three wire planes. Two wires planes, Plane 0 and Plane 1, have angles $\pm$60 degrees from the third plane, Plane 2. Plane 0 and Plane 1 measure ionized electrons in the argon using the induced signal of electrons traveling towards and away from these induction wire planes. The drifting electrons end on Plane 2, which reads a signal from the ionized electrons by collecting them on the wire."
- Figure 3: It is mentioned in the caption that the modified hit is shown, while the original hits only are shown as well as the original and modified waveforms as follows from the legend. If my understanding is correct, the more clear beginning of the caption might be: "Examples of original and modified waveforms and original hit time distributions for single hit (left) and two ...."
- Now says, "Examples of the original and modified waveforms for a single hit (left) and two concurrent hits from a cosmic ray muon overlaid in simulation and a simulated physics interaction (right). Only the waveform from the simulated physics event is modulated to form the new hit in the varied sample. The images come from \cite{wiremod}."
- Line 74: The mean time t0 is not present in the following formula 1.
- Fixed now says: "Equation~\ref{eqn:wireMod} shows the reweighting function as a function of the drift time in terms of the mean time ($t$), width ($\sigma$), and charge (Q) between the original hit ($\sigma$, Q) and the reweighted hit ($\sigma$', Q') for times associated with the waveform being varied ($t_j$)."
- Line 75: Typo: "sigma'" should be corrected in Latex.
- DONE
- Line 92: Possibly omitted verb. As a LArTPC on the surface --> As a LArTPC stays on the surface
- Now says, "Since MicroBooNE operates on the surface, the cosmic ray muon flux is so significant that an abundance of ionized electrons and positively charged argon ions perpetually exist in the detector."
- Equation 2: What does subscript "i" corresponds to? Should be clarified.
- Now says, "These cosmic ray muons samples are used to smear the dQ/dx calibration as a function of position. The smearing function (C) for a generic position variable (i), such as YZ or X, is seen in Equation~\ref{eqn:c} in terms of the global median dQ/dx and the local dQ/dx."
- Furthermore, to keep the position labels consistent in terms of cases I change the cases in the last paragraph to say, "... as a function of drift distance (X), height (Y), distance across the length of the detector (Z), and angles ($\theta_{XZ}$, $\theta_{YZ}$). These hits come from a Gaussian fit to regions of interest on the waveform~\cite{wiremod}."
- Equation 4: The Cx and Cyz are not defined. If I understood correctly, they should be somehow associated with eq.2: e.g. C_{i,YZ} and C_{i,X}.
- Now says, "These cosmic ray muons samples are used to smear the dQ/dx calibration as a function of position. The smearing function (C) for a generic position variable (i), such as YZ or X, is seen in Equation~\ref{eqn:c} in terms of the global median dQ/dx and the local dQ/dx."
- Figure 6 caption: and the track range --> and the energy calculated from the track range
- DONE
- Line 137: Typo: Landua --> Landau
- DONE
- Figure 8: The legend notations should be clarified in the caption (or in the text body): what are "CC" and numbers shown in parentheses (weighted numbers of events in mc?).
- Now says, "Total electron energy reconstructed from selected stopping muons using charged current (CC) muon neutrino events. Statistics for the simulation are scaled to the beam protons on target used in the data distribution. Plot from~\cite{michel_2021}."
- Figure 9: The legend notations should be clarified in the caption (or in the text body): what are "NC", "CC", "OffVtx", Background and numbers shown in parentheses.
- Now says, "Total reconstructed neutral pion mass from charged current (CC) events, neutral current (NC) events, and neutrino interaction events off vertex (OffVtx). Statistics in simulation scaled to beam protons on target used for data distribution. Figure originally from~\cite{michel_2021}."
- Found typo where it says electron instead of neutral pion mass, fixed this.
Reviewer 2 Report
Dear colleague,
It has been a pleasure to review your article 'Energy reconstruction and calibration of the MicroBooNE LArTPC'. It corresponds very well to your presentation at CALOR, reviewing MicroBooNE signal processing techniques, TPC calorimetry calibration, and examples of application in particle identification and energy reconstruction.
The manuscript is well structured, with proper citations of the relevant papers and judicious choice of figures. The text is mostly very clear, with only few typos and sentences a bit obscure that you will find listed below.
I happily support its publication in the special issue of MDPI Instruments for CALOR proceedings.
Detailed comments with suggestions:
* l 22-24: 'these pages', 'The piece' => maybe use 'proceedings' or 'article' instead ?
* l 41-42: 'eliminating ADC values being measured around the vertex' => I cannot understand what this means. Maybe a part of the
sentence is missing ?
* l 48-49: 'implemented in the simulation' => used twice in the sentence
* l 60-61: 'The simulated cosmic ray muons are taken from reconstructed data and overlaid onto the simulation.' => I cannot
understand this sentence. Can you please expand or rephrase ?
* l 61-63: This sentence that talks about the varied samples should probably go after the sentence on l64 that introduces these
samples.
* l 64: what is unisim ?
* l 75: missing $ around sigma'
* Eq 1: In the exponential on the numerator, I believe sigma should be sigma', while t'j should be tj
* Fig 4: The unit for the z axis (color) is missing
* l 132: 'predicted region between 250-450MeV' => should be 'predicted value in the region between 250 and 450MeV'
* l 137: typo 'Landua'
* Fig 6, caption: 'and the track range' => 'and by the track range'
* l 155: 'show' => 'shower'
* l 161: 'reveal' => 'display', or 'show'
* l 164 165: 'included focused' => one of the verbs should be removed
* References: For Microboone references, it is a bit unusual to cite ~10 authors before putting 'et al.'. Suggestion: cite only
Abratenko, P. et al., or maybe even use only 'The Microboone Collaboration'
Author Response
- l 22-24: 'these pages', 'The piece' => maybe use 'proceedings' or 'article' instead ?
- Done as suggested
- * l 41-42: 'eliminating ADC values being measured around the vertex' => I cannot understand what this means. Maybe a part of the sentence is missing ?
- Sentence now says, "With the implementation of electronics response filtering and calibration, the blue bands in Figure~\ref{fig:signal} disappear, eliminating extraneous reconstructed hits surrounding the candidate neutrino interaction vertex."
- * l 48-49: 'implemented in the simulation' => used twice in the sentence
- Sentence now says, "The wire response simulated depends on the liquid argon property values used."
- * l 60-61: 'The simulated cosmic ray muons are taken from reconstructed data and overlaid onto the simulation.' => I cannot understand this sentence. Can you please expand or rephrase ?
- Paragraph now says, "Figure~\ref{fig:wireModX} shows the ratio for all three wire planes as a function of position across the drift distance (x). The trajectories of the cosmic ray muons simulated are taken from real reconstructed data cosimc ray muons, while all other elements of the simulated muons come from CORSKIA~\cite{corsika}."
- * l 61-63: This sentence that talks about the varied samples should probably go after the sentence on l64 that introduces these samples. * l 64: what is unisim ?
- Lines 61-64 now say, "Second, these ratios go into a function that creates variation simulation samples, one made for each trajectory variable (X, YZ, $\theta_{XZ}$, $\theta_{YZ}$) for a total of four samples. The varied samples change the simulated waveforms associated with simulated neutrino interactions, which will be independent of the cosmic ray muon samples used to generate the ratios."
- Furthermore, to keep the position labels consistent in terms of cases I change the cases in the last paragraph to say, "... as a function of drift distance (X), height (Y), distance across the length of the detector (Z), and angles ($\theta_{XZ}$, $\theta_{YZ}$). These hits come from a Gaussian fit to regions of interest on the waveform~\cite{wiremod}."
- To keep with capitalization and for simplicity, the caption for Figure 2 has been changed to say, "Plots of the ratios between data and simulation as a function of drift distance (X). The anode is approximately at x=0. Images come from \cite{wiremod}."
- To keep with the number of variables discussed, a typo is fixed in the last paragraph in Section 2. It now says four wire modification samples, instead of incorrectly five.
- * l 75: missing $ around sigma'
- Done as suggested
- * Eq 1: In the exponential on the numerator, I believe sigma should be sigma', while t'j should be tj
- Done as suggested
- * Fig 4: The unit for the z axis (color) is missing
- Caption and text now fixes omission. Text says, "Equation~\ref{eqn:c} is first used in terms of the detector height and length (YZ) to form $\mathrm{C_{YZ}}$. The event sample statistics in data for YZ are shown in Figure~\ref{fig:yz}. The corrections in YZ aim to address effects, such as unresponsive channels, space charge effect distortions in YZ, and electronics response. Then, the same process is used for $\mathrm{C_X}$ to calibrates as a function of drift distance (X), which corrects for attenuation due to electronegative impurities, diffusion effects, and remaining space charge effect corrections in the horizontal direction. The final calibration of dQ/dx aims to fix time-dependent differences between data-taking days~\cite{calib} (Equation~\ref{eqn:attn})."
- Caption says, "Display from data of number of hits measured as a function of position of the detector height (Y) and length (Z). The image comes from \cite{calib}."
- Found typo where it says "correction" but this plot actually shows hit statistics, I have altered this onerous mistake.
- * l 132: 'predicted region between 250-450MeV' => should be 'predicted value in the region between 250 and 450MeV'
- Now says, "The calorimetric most probable value for dQ/dx in a bin is compared to the Landau-Vavilov predicted value in the region between 250 and 450 MeV."
- * l 137: typo 'Landua'
- Done as suggested
- * Fig 6, caption: 'and the track range' => 'and by the track range'
- Now says, "Comparison between the total energy of stopping muons measured by hit-by-hit calorimetry and the energy calculated from the track range. Taken from \cite{calib}."
- * l 155: 'show' => 'shower'
- Done as suggested
- * l 161: 'reveal' => 'display', or 'show'
- Done as suggested
- * l 164 165: 'included focused' => one of the verbs should be removed
- used word "focused" only now.
- * References: For Microboone references, it is a bit unusual to cite ~10 authors before putting 'et al.'. Suggestion: cite only Abratenko, P. et al., or maybe even use only 'The Microboone Collaboration'
- Done using Abratenko, P. et. al. Furthermore, the delta signal that is not be compiled has been changed to the word "Delta."
Reviewer 3 Report
Dear Madams/Sirs,
in my humble opinion, the proposed paper is very interesting and its scientific level is very high.
It can be accepted with some minor corrections highlighted in the attached file.
Sincerely,
Gabriele Bigongiari

Author Response
- Change line 16 to "allows"
- Done
- Rephrase repetition in Lines 48-49
- Now says, "The wire response simulated depends on the liquid argon property values used."
- Line 64 says unisim, what does that mean?
- Sentence now says, "Second, these ratios go into a function that creates variation simulation samples, one made for each trajectory variable (X, YZ, $\theta_{XZ}$, $\theta_{YZ}$) for a total of four samples. The varied samples change the simulated waveforms associated with simulated neutrino interactions, which will be independent of the cosmic ray muon samples used to generate the ratios."
- Line 155 should say "shower"
- Fixed, thanks